# Design principles for electronic charge transport in solution-processed vertically stacked 2D perovskite quantum wells

Hsinhan Tsai[1,2], Reza Asadpour [3], Jean-Christophe Blancon[1], Constantinos C. Stoumpos[4,5], Jacky Even [6], Pulickel M. Ajayan[2], Mercouri G. Kanatzidis[4], Muhammad Ashraful Alam[3], Aditya D. Mohite [1,7] & Wanyi Nie [1]

State-of-the-art quantum-well-based devices such as photovoltaics, photodetectors, and light-emission devices are enabled by understanding the nature and the exact mechanism of electronic charge transport. Ruddlesden–Popper phase halide perovskites are two-dimensional solution-processed quantum wells and have recently emerged as highly efficient semiconductors for solar cell approaching 14% in power conversion efficiency. However, further improvements will require an understanding of the charge transport mechanisms, which are currently unknown and further complicated by the presence of strongly bound excitons. Here, we unambiguously determine that dominant photocurrent collection is through electric field-assisted electron–hole pair separation and transport across the potential barriers. This is revealed by in-depth device characterization, coupled with comprehensive device modeling, which can self-consistently reproduce our experimental findings. These findings establish the fundamental guidelines for the molecular and device design for layered 2D perovskite-based photovoltaics and optoelectronic devices, and are relevant for other similar quantum-confined systems.

[1] Division of Materials Physics and Application, Los Alamos National Laboratory, Los Alamos, NM 87545, USA. [2] Material Science and Nano Engineering Department, Rice University, Houston, TX 77005, USA. [3] School of Electrical and Computer Engineering, Purdue University, West Lafayette, IN 47907, USA. [4] Department of Chemistry, Northwestern University, Evanston, IL 60208, USA. [5] Department of Materials Science and Engineering, Northwestern University, Evanston, IL 60208, USA. [6] CNRS Institut FOTON — UMR, 6082, Univ Rennes, INSA Rennes, 35708 Rennes, France. [7] Department of Chemical and Biomolecular Engineering, Rice University, Houston, TX, USA. Correspondence and requests for materials should be addressed to A.D.M. (email: adm4@rice.edu) or to W.N. (email: wanyi@lanl.gov)

Ruddlesden–Popper phase layered perovskites are quantum well structures naturally formed by $n$ blocks of inorganic $[PbI_6]^{2-}$ octahedral slab separated by bulky organic cations (R) following the chemical formula $R_2R'_{n-1}Pb_nI_{3n+1}$, here R' is $CH_3NH_3$. While the organic spacers act as an intrinsic protection and passivation layer against moisture, it also leads to a strong quantum and dielectric confinement, which plays an important role in the optoelectronic properties[1–4]. These confinement effects not only widen the optical band gap relative to the 3D perovskites but also confine electron–hole pairs to form bound excitons with binding energy greater than room temperature[5–9]. Both properties are nonideal for photovoltaic devices, which require strong light absorption that overlaps with the solar spectrum and the separation of electron–hole pairs for photocurrent collection. Until recently, its use as light absorbers for solar cells has been motivated by the breakthrough in the synthesis of layered 2D perovskites with higher $n$ values ($n > 3$)[10–13], with broader light absorption across the solar spectrum[14–18]. This coupled with our capability to grow highly crystalline Ruddlesden–Popper layered perovskite thin films with preferential out-of-plane orientation[16] have enabled photovoltaic cells with power conversion efficiency (PCE) approaching 14% in a simple planar configuration.[15] Moreover, our recent work also elucidated a unique internal charge separation of the optically generated excitons via lower energy states in layered 2D perovskites with $n>2$[19]. However, despite these promising breakthroughs along with the demonstration of technologically relevant environmental and photostability, it has been challenging to achieve PCEs on par with the 3D perovskites. To realize the tremendous potential of layered 2D perovskites for photovoltaics and other efficient optoelectronic devices, it is critical to understand the fundamental physical mechanisms that limit the efficient charge transport in these systems.

In this study, through extensive device characterization and modeling, we elucidate the dominant charge transport mechanism during solar cell operation and identify the key bottlenecks that limit the overall efficiency in layered 2D perovskites. The thickness-dependent device characteristics reveal that while absorption can be enhanced with a thicker layer, the overall performance is then limited by transport. Therefore, planar $p$–$i$–$n$ junction cell efficiency reaches a peak value with 200 nm absorber thickness where photogenerated carrier separation and transport are highly efficient assisted by the strong internal electrical field. However, in sharp contrast to 3D perovskites, the recombination for 2D perovskite device increase significantly once the absorber thickness exceeds the critical value. Light intensity-dependent measurements suggest that photogenerated carriers can be efficiently collected at short circuit (SC) condition while the performance is undermined by bimolecular recombination in the low-field regime. In addition, we show that the electronic transport is thermally activated, suggesting that charge carriers need to surmount potential barriers before they are collected at the contacts. To interpret all the observations, we propose a model based on stacked quantum wells where charge collection occurs through transporting across multiple potential barriers, similar to classical semiconducting quantum well systems[20]. The potential barriers are thought to arise from the presence of imperfect stacking of the inorganic slabs in thin films, which may introduce organic spacers that intermittently disrupt the conducting pathway leading to field-dependent charge collection. Our model self-consistently reproduces our experimentally observed device behavior, thus validating that the key bottleneck limiting the photocurrent collection is indeed field-dependent charge separation through the barriers. Our results provide the fundamental guidelines for the design of layered 2D perovskites for high-efficiency photovoltaic devices, which will require improving

the light absorption, engineering highly doped contacts to facilitate efficient charge separation and collection for thicker films, or investigating the incorporation of conducting (organic or inorganic) spacer molecules to reduce the potential barriers.

## Results

This study focuses on the solar cell characteristics in planar $p$–$i$–$n$ device configuration employing layered 2D $BA_2MA_3Pb_4I_{13}$ material ($Pb_4$ unless otherwise mentioned) as the light-absorbing layer (see Supplementary Note 1 and Supplementary Fig. 1) sandwiched between p-type (poly (3,4-ethylenedioxythiophene-polystyrene sulfonate, PEDOT:PSS) and n-type ([6,6]-phenyl-$C_{61}$-butyric acid methyl ester, PCBM) contact layers (see Fig. 1a for layered 2D perovskite structure and device architecture). To understand the solar cell operation principles, we first vary the absorbing layer thickness and characterize the solar cell performance. Figure 1b shows the current density–voltage ($J$–$V$) curve of the planar cell with various absorber layer thicknesses (from 100 to 620 nm) under air mass 1.5 global (AM 1.5 G) solar simulator illumination with 1-Sun equivalent light intensity (100 mW cm$^{-2}$). Along with the $J$–$V$ curve, external quantum efficiencies (EQEs) for those devices are illustrated in Fig. 1c. The EQE curves indicate an absorption onset value of 1.63 eV (760 nm), which is consistent with the band gap expected for the 2D $Pb_4$ perovskite thin film.[13,21] From the $J$–$V$ and EQE, we observe a strong dependence of the photovoltaic performance on the layered perovskite film thicknesses, strongly reflected in the magnitude and shape of the SC current density curves and consequently the amplitude of the EQE spectrum.

We analyze the results by extracting the PCE, SC current density ($J_{SC}$), and open circuit (OC) voltage ($V_{OC}$) from the light $J$–$V$ curves, and plot them as a function of film thicknesses as illustrated in Fig. 1d–f. As a comparison, the thickness-dependent $J_{SC}$ for 3D methyl ammonium lead triiodide absorber in the same device structure is also plotted in Fig. 1f. The overall PCE of the layered perovskite cell plotted in Fig. 1d increases sharply as the film thickness increases from 120 nm (PCE = 7 ±1.21%) and reaches a maximum value for a thickness of 220 nm (PCE = 11.37 ± 0.97%). This is mainly due to the enhancement in the absorption with increasing film thickness as confirmed by the EQE spectra in Fig. 1c. However, beyond a film thickness of 220 nm, we observe a significant drop in the PCE down to 6.89 ± 1.17% for 450 nm film thickness. Analysis of the $J_{SC}$ as a function of thickness curve (Fig. 1e) exhibits a peak value for a thickness value of 220 nm followed by a monotonic decrease for the thicker films. Such dependence is consistent with the integrated $J_{SC}$ value from the EQE spectrum (Fig. 1c) taken under the SC condition, validating the $J$–$V$ measurement. Comparison of the thickness dependence of the $J_{SC}$ from the 2D and 3D perovskite device in Fig. 1e shows that the $J_{SC}$ for 3D perovskite devices (black curve) is insensitive to the film thickness between 250 and 600 nm range. We further study the different $n$-number ($n = 2$ and 5) compounds along with the film thickness dependence (Supplementary Fig. 2 and Supplementary Note 2). The thickness dependence becomes more pronounced when $n$ is increased and low-field collection becomes less efficient for lower $n$ number (Supplementary Figs. 2 and 3 and Supplementary Note 3).

From the thickness-dependent results, we conclude that charge collection is the most efficient for the optimized film thickness, which leads to high fill factor (FF) value (74.1 ± 2.2%) and $J_{SC}$ (14.92 ± 1.73 mA cm$^{-2}$). High $V_{OC}$ value (1.0±0.05 V) is an indication of low non-radiative recombination at OC. This excellent efficiency in simple planar device architecture is unusual considering the fact that photogenerated carriers in these 2D perovskite systems are excitonic in nature[5–7]. However, once the

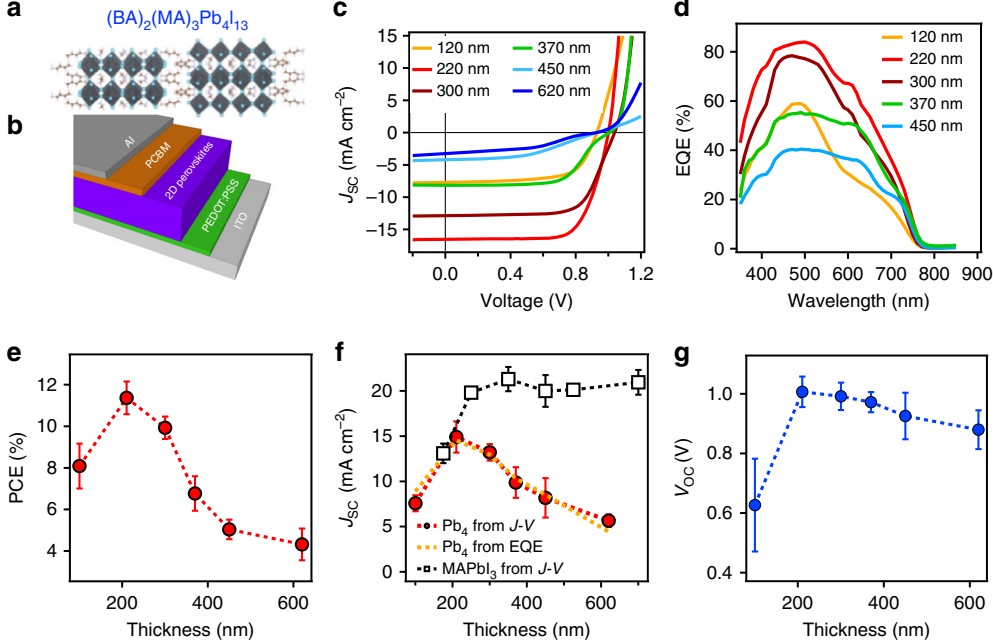

**Fig. 1** Photovoltaics with various absorber thickness evaluated by *J–V* characteristics. **a** Molecular structure of BA$_2$MA$_3$Pb$_4$I$_{13}$ (Pb$_4$) and **b** planar solar cell device structure used in this study. **c** *J–V* curve under AM 1.5 G solar-simulated light with 1-Sun equivalent power and **d** EQE spectrum collected under short-circuit condition with monochromatic light for planar device with various Pb$_4$ film thicknesses. **e**–**g** Extracted average PCE, $J_{SC}$ and $V_{OC}$ as a function of film thickness. The $J_{SC}$ for a device with methyl ammonium lead triiodide (MAPbI$_3$) as an absorber in the same structure is measured as a comparison in **f** (black curve). Error bars in **e**, **f** were s.e.m. collected over eight devices of various absorber thickness for statistics

thickness is above 200 nm, photogenerated carriers decay to ground states before being collected at the contact, leading to inefficient collection that reduces the $J_{SC}$, FF, and thus the PCE. This is also consistent with the trend observed for $V_{OC}$ as a function of thickness in Fig. 1f, which decreases as the film thickness increases. In contrast, the $J_{SC}$ for 3D perovskite cell is less sensitive to the film thickness in the 300–500 nm range because optical excitation produces free carriers[22–24] that can be efficiently collected at the contacts, even for thicker layers as the cell operates in the non-Langevin type of recombination regime.

As a summary, the final performance of layered perovskite solar cell is defined by the competition between photogeneration (absorption) and carrier transport through the layer thickness. A simple increase in film thickness would improve the absorption, but in the mean time reduces the carrier transport, which reduces the overall efficiency.

To gain deeper insight into the carrier transport processes, we examine the *J–V* characteristics under 1-Sun illumination and extract the slope at SC and OC conditions, respectively, at each of the thicknesses (see Fig. 2 and Supplementary Fig. 4). Figure 2a shows the light *J–V* curve normalized by the $J_{SC}$ value for devices with thicknesses ranging from 220 to 620 nm. To quantify the charge collection, we take the slope of the *J–V* curves at each voltage (Supplementary Fig. 4) and plot the values near SC (high field) and OC (low field) as a function of film thicknesses as shown in Fig. 2b. The slopes from *J–V* curves at OC and SC are generally interpreted in terms of extrinsic series or shunt resistances, respectively. However, an analysis of the dark *J–V* curve shows that the contributions from extrinsic resistances are small (Supplementary Figs. 5 and 6). We therefore take the slopes as an indication of charge collection efficiency as a function of internal field[8,25–27], a presumption supported by the device simulation described later in the paper. In an ideal case, at open circuit conditions (OC), a nearly vertically steep slope value represents a fast transition from maximum power point (MPP) to OC, indicating that the current collection is efficient even at low field,

However, a gradually changing slope value means that the device loses photocurrent when approaching the low-field regime. On the other hand, when the internal field reaches the maximum at SC, a flat (horizontal) *J–V* slope is expected, whereas a steep slope at this regime suggests that the charge collection is strongly field dependent.

Employing such analysis on the *J–V* data in Fig. 2a, we find that the device with optimum thickness 220–300 nm presents a near constant slope at SC and becomes steep at OC in Fig. 2b, indicating that the charges can be extracted efficiently at both high- and low-field regime when illuminated under 1-Sun power. For a device with absorber thickness >300 nm, the slope near OC gets reduced as observed from the *J–V* curve (see Fig. 2a); this reflects a sharp drop in slope value at OC in Fig. 2b, while that at SC remains flat. This indicates that charge collection efficiency is more sensitive at OC when the film is thicker. For device thickness above 400 nm, the field-dependent charge collection gets stronger at both the conditions. To confirm that the orientation and packing of the inorganic slab does not vary substantially with increasing thickness, we present grazing incidence-wide angle X-ray scattering (GIWAXS) maps to examine Pb$_4$ thin films with 220 and 450 nm thicknesses in Fig. 2c, d. The results show that both films have near-identical crystallinity and preferential out-of-plane orientation, as evidenced by the discrete Bragg spots and strong (111) peak along $q_z$ direction[16].

The increased absorber thickness leads to two changes in the system; first, the distance between two electrodes is enlarged (while orientation remains similar), and second, the net field that drops across the film is reduced (Supplementary Note 4, Supplementary Figs. 8–9, and Supplementary Tables 1–4 for simulated field profile for higher absorber thicknesses and simulation method). At SC, the charge collection relies on electron–hole pair separation followed by transport drifting by the internal field. In contrast, at OC, the net field is low and the collection relies on carrier diffusion, and any potential barriers (insulator or trap states) will significantly reduce the collection efficiency. In our

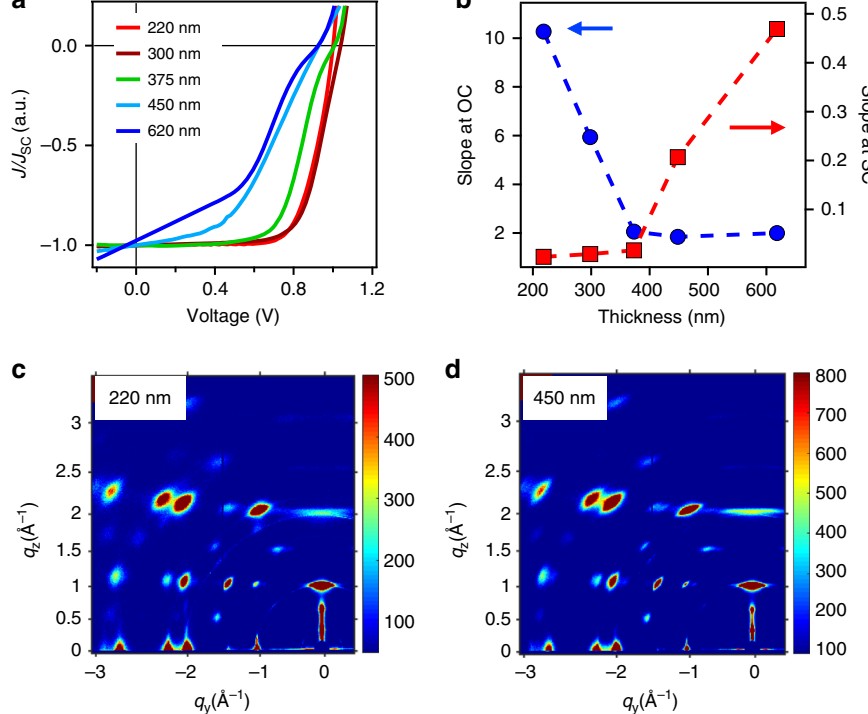

**Fig. 2** *J–V* characteristic curves for solar cells with different absorber thicknesses. **a** *J–V* curves normalized by $J_{SC}$ for various thicknesses from Fig. 1b. **b** The *J–V* slope obtained by taking the first-order derivative from **a** at OC (red) and SC (blue). **c**, **d** GIWAXS map for Pb$_4$ thin film at 220 and 450 nm, respectively

case specifically, the photocurrent collection for layered perovskite solar cells with a thickness of 220 nm is highly efficient resulting in a high FF. Beyond 300 nm, the carrier diffusion at OC is greatly limited by the large distance and small field between hole-transporting layer (HTL)-selective and electron-transporting layer (ETL)-selective contacts, which increases the probability of carrier recombination losses. This is because layered perovskite is an intrinsic semiconductor with low doping density[16] and the depletion region width is 200–300 nm, beyond which a flattened electrical field profile occurs. This thickness is thus the upper limit for the field drop across the device, as determined by device simulations (Supplementary Figs. 7–9), and we consider that the field decreases monotonically from the HTL side to the ETL side in this device configuration (Supplementary Fig. 10). Therefore, the photogenerated carrier collection, especially at the center of the film for the layered perovskite system, is strongly field dependent, leading to a reduction in the *J–V* slope near OC, while the SC slope remains flat. For film thickness exceeding 500 nm, both drift and diffusion are inefficient, which essentially lowers the photocurrent collection in both conditions.

To understand the exact loss mechanism at SC and OC, we performed light intensity-dependent *J–V* characteristics for the layered perovskite device with two thicknesses as shown in Fig. 3. These experiments allow probing the absorption and transporting trade-off for charge collection through an energy landscape dominated by quantum wells independently. Figure 3a, b shows the *J* (normalized to $J_{SC}$) as a function of effective internal field ($V–V_{OC}$) under various light intensities from 10 to 500 mW cm$^{-2}$ for thickness of 220 and 375 nm devices, respectively. The extracted $J_{SC}$ and $V_{OC}$ as a function of light power are plotted in Fig. 3c, d. In Fig. 3a, b, we found that the slopes near SC for both thicknesses do not vary. The $J_{SC}$ values as a function of light intensity follow an almost linear dependence in the power range of 1–500 mW cm$^{-2}$ as shown in Fig. 3c. Both these data suggest that the charge collection efficiency or charge recombination loss

is independent of incident light intensity (carrier density) at SC[27,28]. In both cases, the bimolecular process, which is highly carrier-density dependent, does not contribute to the photocurrent loss at SC, where all the free carriers can be collected due to the presence of a strong internal electric field[27,29,30]. The majority loss should be through a monomolecular process from either Coulombically bound electron–hole pair recombination or a trap-assisted process[28,30,31]. In contrast, the slope near a low field (from MPP to OC) changes most dramatically for higher light intensities. This is attributed to the enhanced bimolecular recombination at low fields and higher carrier densities[27,29,30,32]. For a 375-nm-thick film, the change in slope occurs at a lower threshold than that in 220 nm film with varying light intensity (Fig. 3b). This is because both the enlarged absorption and the higher probability of bimolecular recombination in thicker films as carriers must diffuse through a larger distance to be collected.

Next, we plot $V_{OC}$ against natural log of light power in Fig. 3d to investigate the carrier recombination losses near OC. The data reveal that the $V_{OC}$ follows two linear regimes, one at high power with a slope of about $1k_BT/q$ and the other near low light with a slope of about $2k_BT/q$ (where $k_B$ is Boltzmann constant, *T* refers to temperature, and *q* is elementary charge)[31]. The thicker film again shows a much lower threshold for the change in slope of $V_{OC}$. The linear dependence is indicative of the recombination process near OC. It first follows a monomolecular process at lower light intensities but then switches to a bimolecular process. This suggests that at lower light powers, the quasi-Fermi-level splitting is reduced with carrier density and consequently decreases the built-in field.[31,33–35] This is also reflected as an enhanced field-dependent current collection near SC under low power (Supplementary Fig. 4), which is associated with the reduction in the built-in field[36]. We also note that dark shunt current is two orders of magnitude lower than the photocurrent even under lowest illumination and does not affect the above analysis (Supplementary Figs. 4 and 5).

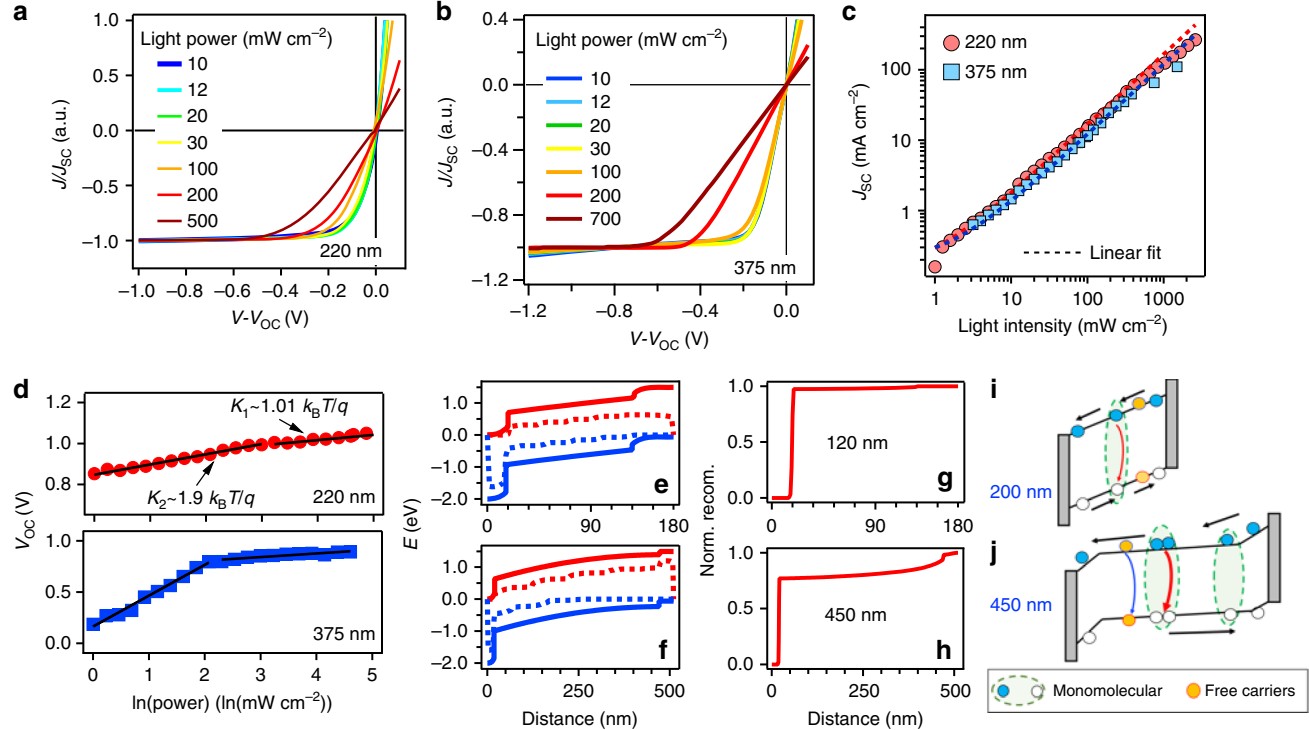

**Fig. 3** Light intensity dependence with device J–V characteristics. Normalized light J–V curves for a device with absorber thicknesses of **a** 220 nm and **b** 375 nm under various illumination intensities. **c** $J_{SC}$ value as a function of light power over a wide illumination range. **d** $V_{OC}$ as a function of illumination intensities for two device thicknesses. **e**, **f** Calculated band energy as a function of film thickness and **g**, **h** normalized recombination profile as a function of distance in the device for two typical thicknesses. **i**, **j** Schematics illustrating the carrier recombination processes in thin film and thick film in the quantum well-layered perovskite structure

The recombination analysis is further supported by device simulation with calculated band diagram in the layered perovskite device and the recombination profile based on the charge density distribution (see Supplementary Note 4 and Supplementary Figs. 8 and 9 for simulation details), as shown in Fig. 3e–h. The band diagram (Fig. 3e, f) clearly shows that the internal electrical field drop at SC condition through a thin device is uniformly strong throughout the film, while that for a thicker layer is reduced, especially at the center of the film, featuring with a flatter region. Such field drop at a thicker layer will promote recombination, and therefore the recombination profile as a function of distance differs dramatically in the two cases (Fig. 3g, h). In the thin layer, the recombination majorly occurs near the interface because the film is fully depleted by the internal field. However, when the film grows thicker, the recombination occurs both at the bulk and near the interface due to the weakened field near the center.

To assist understanding over the discussion above, we draw a schematic illustration that describes the different recombination processes during the planar cell operation in Fig. 3i, j. For thin layers (in 200–350 nm range) when the field is strong across the device, the bound carriers can be separated into free carriers (Fig. 3i), which are subsequently collected by the strong internal field at SC. The major loss mechanism is diffusion-limited free carrier recombination at low field (near OC), and larger thicknesses will lead to more free carrier recombination loss. However, the free carriers can be collected near SC quite efficiently, as evidenced by the linear relationship of $J_{SC}$ versus power curve within this thickness range, and the reduction in $J_{SC}$ from 200 to 375 nm is thus attributed to monomolecular loss. When the film thickness increases to 450 nm (Fig. 3j), the internal field drop is greatly reduced especially near the center of the film. The photogenerated carriers can only undergo a partial separation, and

non-separated carriers generated in the center of the film recombine strongly through a monomolecular process, which results in strong field dependence near SC as well as OC. Such picture agrees with the recombination analysis based on the $V_{OC}$ as a function of light intensity curves for both the thicknesses in Fig. 3d and the recombination profile in Fig. 3e–h.

To differentiate possible origins leading to the monomolecular recombination, we conducted device J–V characteristics under low temperatures that are presented in Fig. 4. The J–V curves plotted in Fig. 4a are normalized by $J_{SC}$ values to compare the field-dependent behavior from SC to MPP. As temperature reduces, the slope near SC is greatly enhanced; this suggests an emergence of field dependence in the photocurrent collection[9,27]. The $J_{SC}$ and $V_{OC}$ values as a function of temperature extracted from the J–V curves are also plotted in Fig. 4b, c. When the temperature decreased from 300 to 200 K, the device loses $J_{SC}$ and FF monotonically as expected for inefficient charge collection (see Fig. 4c, d). The $V_{OC}$ first increases as the temperature is lowered to 260 K, which is consistent with lowered dark injection (recombination) current at low temperature[27,37]. Further, the $V_{OC}$ tends to saturate as the temperature is further lowered to around 200 K, most likely due to the competition between the suppressed thermal recombination and enhanced bimolecular recombination, which otherwise should continue to increase linearly with T. To quantify the collection efficiency with temperature near SC, we plot the slope of J–V curve as a function of T in Fig. 4d. The slope decreases exponentially with temperature, which is expected from a classical temperature dependence described using a thermally activated model. The inverse slope $\ln (d(J/J_{SC})/dV)^{-1})$ against $1/T$ is plotted in Fig. 4e following Arrhenius equation to obtain an activation energy ($E_a$). From the linear fit, the $E_a$ is estimated to be around 130 meV (in temperature range of 300–200 K).

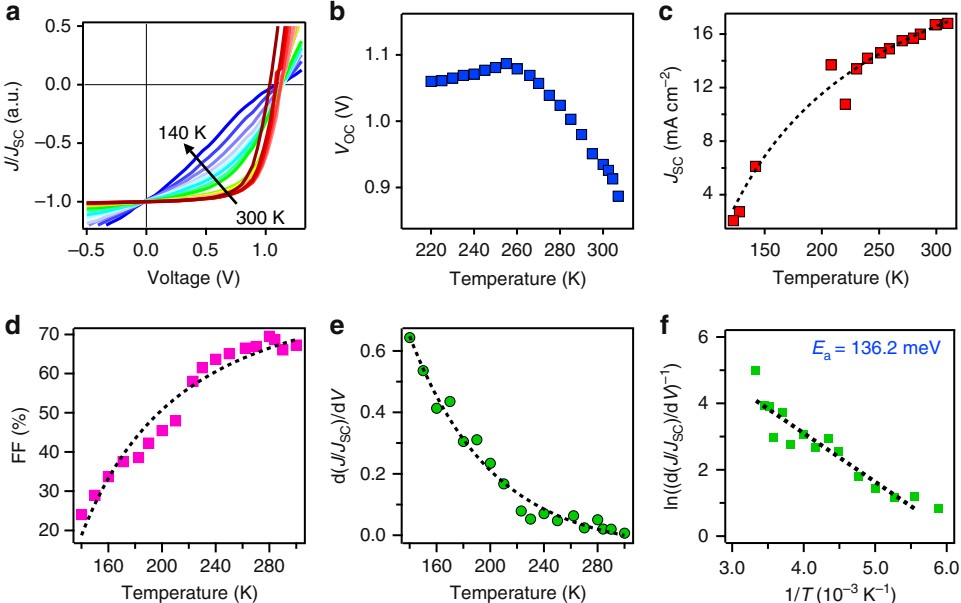

**Fig. 4** Device characterization under low temperatures. **a** Normalized light $J$–$V$ curves under 1-Sun illumination for a device with absorber thickness of 220 nm. **b** $J_{SC}$, **c** $V_{OC}$, and **d** fill factor (FF) as a function of temperature with the dashed line as guidance to the eye. **e** Extracted $J$–$V$ slope at SC at various temperatures with the dashed line as fitting curve and **f** the linear plot following Arrhenius equation by plotting the inverse $J$–$V$ slope in natural log scale against $1/T$ with dashed line as the linear fit

The emergence of $J$–$V$ slope near both SC and OC indicates the presence of energy barriers for photocurrent collection, intrinsically existing in the thin device, and is more pronounced at lower temperature. As the temperature gets lower, the carriers lose their thermal energy, thus making it harder for carriers to surmount the barrier in spite of the fact that the barrier height remains unchanged. This results in the loss of charge carriers in the process of hopping across the potential barriers, which decreases the overall photocurrent.

Finally, to consistently interpret all the experimental observed data from three independent measurements, we propose a model specially designed for the 2D perovskite system where a vertically stacked quantum well is present throughout the film thickness. Based on that, we conduct theoretical device modeling to interpret all the data consistently as demonstrated in Fig. 5. Figure 5a schematically illustrates the energy band landscaped, reflecting the fact in a layered 2D perovskite quantum well system. Unlike the conventional quantum wells where wide band gap materials are stacked in a layer-by-layer fashion out of plane, in our case, those quantum wells are stacked vertically as demonstrated by our previous work[2]. Even though the thin films were grown with preferred vertical out-of-plane direction, the slight mismatch and imperfect crystal packing in thin films (in comparison to single crystals) manifests as potential barriers created by the partially intercepting organic spacers between conducting inorganic slabs that could interrupt the transport pathway (Fig. 5i). This band diagram thus identifies two types of carriers: quasi-bound carriers localized in low-energy levels in inorganic regions and free delocalized carriers outside the potential barriers. The electrons and holes are repeatedly trapped into and detrapped from the quasi-bound states as they move toward their respective contacts.

Based on this model, the experimental $J$–$V$ characteristic trends are reproduced by simulation as shown in Fig. 5b–h; the calculated band diagram with multiple potential barriers is shown in Supplementary Note 4 and Supplementary Figs. 11 and 12. For example, Fig. 5b, c explains the thickness-dependent turn around of the photocurrent shown in Fig. 1b, e. For very thin absorbers

(<200 nm), the photoabsorption is incomplete (Supplementary Note 6 and Supplementary Fig. 13); therefore, even though the high internal field ($E \propto V_{bi} W^{-1}$) assists in the collection of most of the photogenerated carriers, resulting in high internal quantum efficiency that is comparable to the best-performing device, the SC current is still low. Here, $V_{bi}$ is the built-in voltage and $W$ is the absorber thickness of the solar cell. For a thicker absorber (over 200 nm), the absorption is essentially complete, but the internal field is now too weak to extract the free carriers through the energy-landscape dominated by transport through trapping and detrapping repeatedly. Consistent with Fig. 1, the photocurrent is maximized at $W = 200$ nm when the photoabsorption is balanced by photogeneration and charge transport. In addition, the $J$–$V$ characteristics in Fig. 5b also show that the evolution of differential conductance (as a function of $W$) reported in Fig. 2 is an intrinsic feature of charge collection through such complex energy landscape and is unrelated to series and shunt resistances of typical solar cells.

Similarly, the same model is employed to predict the light intensity and temperature-dependent plots as summarized in Fig. 5d, e. The result provides consistent validation of the experimental data reported in Figs. 3 and 4, respectively. The linear increase in $J_{SC}$ with intensity shown in Fig. 3c is intuitively obvious. More interesting, Fig. 5d explains the non-intuitive reduction in FF (as seen in Fig. 3b) as a consequence of increased recombination of quasi-bound carriers at higher intensity. Finally, Fig. 5f–h explains the reduction of $J_{SC}$, $V_{OC}$, and FF with temperature. Lowering the temperature decreases the thermal energy of the carriers trapped within the energy wells: the carrier cannot escape as easily, and thus the recombination is greatly enhanced. Beyond 280–300 K, the carriers have sufficient energy to surmount the energy barrier, and thus $J_{SC}$ begins to saturate beyond the temperature. On the other hand, the FF of the device drops drastically with temperature (Fig. 5h), indicating a thermally activated charge collection in the system. Consistent with the activation energy of photocurrent collection following Arrhenius relationship of $J$–$V$ slope against $1/T$ in Fig. 4e, the barriers between the regions differ by 150–200 meV from device

simulation. We acknowledge that the relative band gap value may change with temperature, which can alter the absolute magnitude of solar cell figure-of-merits. However, the trend for carrier transport under low temperature does not change. This is supported by the trend by comparing the experimental data to simulated curves on $J_{SC}$ and FF (Fig. 5f, h), because the relative barrier height remains invariant (Supplementary Figs. 11 and 12), which is the dominant factor that influences the carrier transport. We also notice that a difference between simulation and experimental data is observed in Fig. 5g for the temperature-dependent $V_{OC}$ curve. This can be attributed to the band gap energy and charge mobility changes at lower temperature. Based on these two factors, the refined model can reproduce the experimental observed trend in $V_{OC}$ (Supplementary Note 7 and Supplementary Fig. 14). However, we emphasize that such refinement in the model will not alter the observed trends.

## Discussion

Based on the above results and discussions, we establish that the key bottleneck in the layered perovskite quantum well photovoltaic devices is the field-limited carrier collection. Such

limitation arises from the presence of multiple potential wells, which requires a strong built-in electric field for thicker 2D perovskite films. Independent of whether the photogenerated carrier is an exciton or free carrier, a strong electric field is necessary for the carriers to surmount the potential barrier before they recombine. In our analysis, the magnitude of the potential barrier incorporates any effects associated with the exciton binding energy. Because such a phenomenological approach is validated by the fact that the model can self-consistently explain three independent experimental data (i.e., intensity, field dependence, and temperature dependence), we can propose the mechanism without explicitly invoking the exact nature of carriers.

In summary, based on our results, we have identified the key bottleneck for charge transport for vertically stacked layered perovskite quantum well photovoltaic devices arising from recombination losses of charge carriers across potential barriers created. These findings present opportunities for improved design of 2D perovskite structures, where long-range vertical packing for facilitating conducting pathways can be achieved that results in reduced numbers of potential barriers. Alternatively, doping the organic spacers to reduce the dielectric contrast could

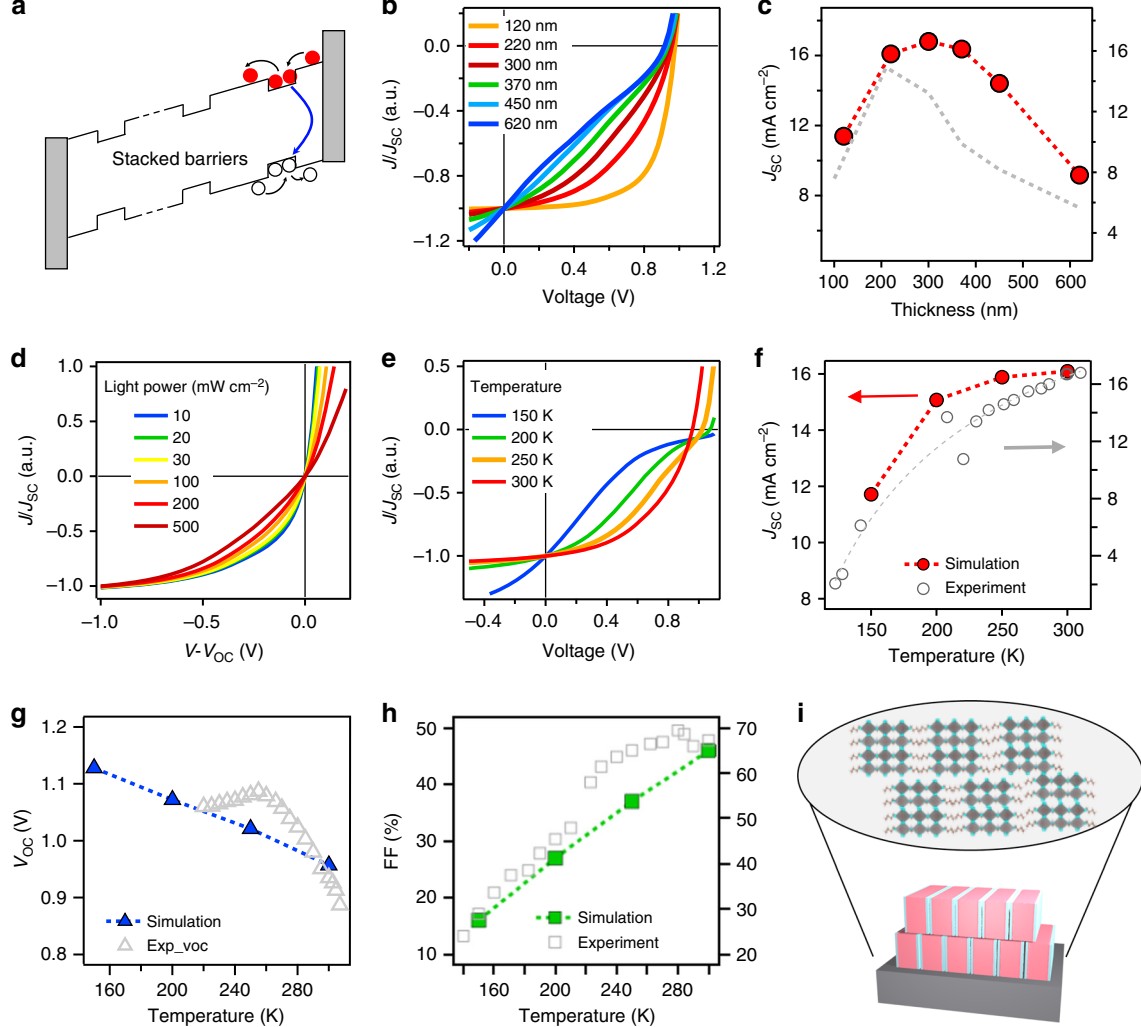

**Fig. 5** Model and simulation results. **a** The model shows potential wells that can prevent carriers from escaping. **b**, **c** The thickness dependence of J–V curves and $J_{SC}$. Very thick structures increase the recombination in wells and reduce $J_{SC}$. **d** Intensity dependence of normalized J–V curves. **e–h** Temperature dependence of J–V curves and the extracted $J_{SC}$, $V_{OC}$, and FF as a function of temperature. The experimental data taken from Figs. 2 and 4 in gray symbols are plotted along with the simulated data to directly compare the two. **i**, schematic illustration of vertically packed quantum wells with misalignment in molecular structures

result in lowering the potential barriers for efficient charge transport. We anticipate that our work will lead the next steps for the incorporation of such strategies, which could overcome this fundamental bottleneck for transport in 2D perovskites and lead to high-efficiency photovoltaics and other optoelectronic devices.

## Methods

**Materials**. Layered perovskite crystals are synthesized according to previously published work[10,12,13]. The obtained crystals are taken for X-ray diffraction characterization to confirm the purity of the compound (Supplementary Fig. 1). Precursors are prepared by dissolving $Pb_4$ crystals in anhydrous N,N-dimethyl-formamide solvent with various molar concentrations (0.112, 0.225, 0.35, 0.45, and 0.6 M) with respect to $Pb^{2+}$ ion. The precursors were stirred for 24 h before using.

**Device fabrication**. Indium tin oxide-coated glass slides were cleaned through standard steps by ultrasonication bath with distilled water, acetone, and isopropyl alcohol for 30 min, respectively. The substrates were then dried under nitrogen flow and treated with oxygen plasma for 3 min. The cleaned substrates were then used for poly(3,4-ethylenedioxythiophene) polystyrene sulfonate (PEDOT:PSS low conducting grade, Sigma-Aldrich) coating. After 5000 rpm for 45 s, spin coating the PEDOT:PSS layer formed in thickness of around 50 nm serves as HTL. After drying at 130 °C for 30 min, all the substrates were transferred into argon-filled glovebox for perovskite layer deposition. We followed our previously developed fabrication procedure[16,38–42], first preheat the substrate at 110 °C for 5 min and then quickly transfer to the spin-coating chunk; 100 μl of previously prepared precursors was then dropped onto the hot substrate and the spin coater started immediately to prevent temperature from quenching. The ETL was done by spin coating PCBM (Nano-C with purity over 99.5%) solution at 1000 rpm for 45 s. Finally, the coated substrates were mounted into vacuum chamber inside the glovebox for aluminum deposition. The film thicknesses were determined by stylus profilometer.

**Device characterization**. All the devices were encapsulated inside the glovebox with glass slide sealed with UV-curable epoxy to prevent from air degradation. The current–voltage characteristics were taken by Keithley 4200 unit under standard solar simulator light source with air mass 1.5 G filter. The intensity was fixed at 100 mW cm$^{-2}$ calibrated by standard silicon diode. The light intensity was varied by focusing the light with lens in combination with neutral optical density filters.

The temperature-dependent characterization was done by mounting the non-encapsulated device in a cryostat chamber with electrical connections. The chamber was pumped down to $10^{-4}$ Torr vacuum and cooled by liquid nitrogen.

**Data availability**. All the data that support the findings of this study are directly available from the authors on request.

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

## Acknowledgements

The work at Los Alamos National Laboratory (LANL) was supported by the LANL LDRD program under grant 20180026DR (XWPG) (W.N., H.T., and J.-C.B.). A.D.M. acknowledges support by Office of Energy Efficiency and Renewable Energy grant DE-FOA-0001647-1544 for this work. The work at Purdue University was supported by the National Science Foundation under Grant No. 1724728. M.G.K. acknowledges the support from the Office of Naval Research grant N00014-17-1-2231 (stability of 2D perovskites). The GIWAXS maps were done with the help of Dr. Joseph W. Strzalka (X-Ray Science Division) and the use of sector 8-IDE in Advanced Photon Source was supported by US Department of Energy (DOE) Office of Science User Facility operated for the DOE Office of Science by Argonne National Laboratory under Contract No. DE-AC02-06CH11357.

## Author contributions

W.N. and A.D.M. conceived the project and direct the experiment. H.T. and C.C.S. synthesized the material and conducted structural characterization. H.T. and W.N. fabricated and characterized the photovoltaic device and analyzed the data. R.A. built the device model and conducted device simulation under the supervision of M.A.A. M. G. K. contributed to the design and selection of materials and to the interpretation of results. All the authors were involved in result discussion and manuscript writing.

## Additional information

**Competing interests:** The authors declare that there are no competing interests.

