## [Peer Review File · Nature Communications]

Reviewers' Comments:

Reviewer #1:

Remarks to the Author:

In this manuscript entitled "Design principles for electronic charge transport in solution-processed vertically stacked 2D perovskite quantum wells", the authors carry out in-depth investigations of how exciton separate and transport across the film with various film thickness. Overall the work presented here is interesting and well-written. Importantly the author found a decay of internal electrical field in thicker film, which is the main cause for the difficulty of extracting carriers. However there are still some points confusing to prove the authors' viewpoints. I would like to recommend accept for publication in Nature Communications once the following issues are properly addressed:

1. The authors proposed a flatten internal electrical field in the thicker film. It seems to be consistent with experiment and device simulation work, will it also appear in 3D film with thicker thickness?
2. What is the main cause for this appearance and how this will be when increasing the n values for the layered perovskite?
3. What is the limit in terms of film thickness for the appearance for this flatten internal field in these layered perovskite? It would be helpful for readers to better understand the story when more explanation is provided.
4. Since the quantum wells are oriented perpendicularly respect to the substrate as depicted by GIWAXS, how the "increased number of organic spacer will intermittently disrupt the conducting pathway" as claimed in the Abstract? This seems to be contradicting.
5. The manuscript stated that "Ruddlesden-Popper halide perovskites are two-dimensional solution-processed quantum wells and have recently emerged as highly efficient semiconductor for solar cell approaching 13% in power conversion efficiency." However, I remember there was a paper on recent EES reported much higher efficiency as high as 13.7%. It should be corrected with proper references to show the state of the art.

Reviewer #2:

Remarks to the Author:

This works provides a comprehensive analysis of 2D perovskite solar cells losses. The best performances are similar to the ones previously reported, and the thickness effect follows an expected trend. However, the presented results can help understanding the working mechanisms of 2D perovskite solar cells.

As the authors mention, the Voc Vs light intensity analysis (Fig 3) suggests a dominating monomolecular recombination process at lower light intensities. Signs of the type of carrier dominating this process could be helpful to further optimization of the system.

In line with the previous authors' report (H. Tsai et al., Nature, 536, 7616, 312–316, 2016), which indicated a charge carrier concentration of $\sim 10^{16} \text{ cm}^{-3}$. If that is the case, the interpretation of the electric field distribution should be discussed, as it could be confined in a depletion region for films $> 200 \text{ nm}$.

Another point which, in my opinion, deserves further analysis is the role of the electric field in the exciton dissociation mechanisms ("when the field is strong across device, the bound carriers can be separated into free carriers"). Discussing the exciton binding for this materials energy could help.

The authors claim "For very thin absorbers ($< 200 \text{ nm}$), the photo- absorption is incomplete, therefore even though the high internal field ($E \propto V_{bi}/W$) successfully collects most of the photo-

generated carriers, the short circuit current is still low". This is reasonable, but could be further confirmed with internal quantum efficiency measurements.

If these issues are addressed, I would recommend the publication of this work in Nature Communications.

Reviewer #3:

Remarks to the Author:

The authors present a study on the charge transport and recombination in 2D perovskite solar cells based on quantum wells. By using JV measurements and simulations they propose a model for the workings of vertically stacked 2D perovskite solar cells. The manuscript represents a nice combination of experimental and simulation work on an important topic.

A few issues:

- 1) The manuscript contains a few typos and other language mistakes that could easily be fixed.
- 2) There is no direct comparison between the experimental data and the simulations. This makes it difficult to assess how good the agreement is. Moreover, the experimental data (fig 4) don't include the FF versus T while the simulations do (fig 5). The latter, however, don't include the Voc. It would be great, if there was a direct comparison between simulations and experiments even if perfect quantitative agreement is not obtained.
- 3) The simulations appear to be well done. However, some details are missing: In the SI it is mentioned that the optical constants are either taken from the literature (which is fine of course) or measured by the authors. In the latter case, the data should also be included in the SI (i.e. n & k for the materials). From table S1 (equations) it is not obvious to me how the offsets in the conduction and valence band (both 130 meV) are implemented. Such details are needed in order to be able to reproduce the results.
- 4) How are the layers implemented in the simulations? For example, how many layers are there and how thick are they? In Figs S6 and S7 the band diagrams seem to show the effect of the variations in conduction and valence band, but it is not clear whether this is actually the case. Also, in the main text, right before the summary, the authors suggest that the density of potential barriers increases with thickness? why? shouldn't the density remain constant?
- 5) Just below fig. 4 the authors state that "As the temperature gets lower, the barrier height increases...". I suppose the authors mean that it is increasingly more difficult to cross the barrier at lower temperatures while the barrier itself is constant. If not, could the authors clarify why the barrier depends on temperature?

We appreciated the reviewers' time and effort for providing the detailed comments and suggestions for this manuscript. We feel with these modifications have greatly improved our MS. Below is a point to point response to each reviewers' comments along with the appropriate revision to the manuscript:

Reviewer #1:

In this manuscript entitled “Design principles for electronic charge transport in solution-processed vertically stacked 2D perovskite quantum wells”, the authors carry out in-depth investigations of how exciton separate and transport across the film with various film thickness. Overall the work presented here is interesting and well-written. Importantly the author found a decay of internal electrical field in thicker film, which is the main cause for the difficulty of extracting carriers. However, there are still some points confusing to prove the authors' viewpoints. I would like to recommend accept for publication in Nature Communications once the following issues are properly addressed:

We are greatly encouraged by the reviewer's overall assessment of our MS and for the specific comments to improve our paper. We have modified the discussion in the manuscript to clarify the points based on the reviewer's suggestions. The detailed response to each point and the related modification in the revised manuscript are attached below.

1) The authors proposed a flatten internal electrical field in the thicker film. It seems to be consistent with experiment and device simulation work, will it also appear in 3D film with thicker thickness?

We agree with the reviewer that the flattening of the internal field is an important aspect of the paper. The flattening arises because one needs to balance photo-absorption (thicker, low-doped material is better) vs. charge collection (thinner, higher doped material is desired). Given typical doping densities, most planar solar cells (e.g. Si HIT [DOI: [10.1002/pip.2959](https://doi.org/10.1002/pip.2959)], CdTe [DOI: [10.1109/PVSC.2015.7355778](https://doi.org/10.1109/PVSC.2015.7355778)], CIGS [DOI: [10.1109/JPHOTOV.2016.2583790](https://doi.org/10.1109/JPHOTOV.2016.2583790)], 3D perovskites [DOI: [10.1109/JPHOTOV.2015.2451000](https://doi.org/10.1109/JPHOTOV.2015.2451000); DOI: [10.1021/acsnano.7b06294](https://doi.org/10.1021/acsnano.7b06294); DOI: [10.1126/science.aaa0472](https://doi.org/10.1126/science.aaa0472)], etc.) are characterized by a flattened internal field region beyond a transition thickness. This transition thickness depends on the doping, mobility, band-profile and the complex dielectric constant of the solar absorber.

Specifically, in our case, the electrical field distribution for our system in the 2D device is based on the charge density distribution calculated from the fitting of the J - V characteristics where a flattened electrical field can be found for a thick device. Similarly, in a 3D perovskite device, the electrical field drop will follow a similar trend. This is supported by both experimental study and theoretical device simulations [DOI: [10.1109/JPHOTOV.2015.2451000](https://doi.org/10.1109/JPHOTOV.2015.2451000); DOI:

[10.1021/acsnano.7b06294](https://doi.org/10.1021/acsnano.7b06294); DOI: [10.1126/science.aaa0472](https://doi.org/10.1126/science.aaa0472), DOI [10.1007/s40820-017-0159-z](https://doi.org/10.1007/s40820-017-0159-z)].

The absolute value will certainly vary depending on the dielectric constant of the 3D perovskite layer, the doping of the electron transport layer and hole transport layer as well as their relative band position. However, because the carrier mobility and diffusion length in the 3D perovskite film is large, the charge collection efficiency remains efficient even for thicker layers, even though a flattened field can be reached in the middle of the film. Thus, the impact of electrical field in the 3D perovskite film is not as appreciable as that in the 2D perovskite device.

2) What is the main cause for this appearance and how this will be when increasing the n values for the layered perovskite?

The reviewer has raised an important aspect that is worth investigating. The internal electric field is determined by two factors: work-function difference between the two contacts and the doping density of the perovskite layers. The electric field transition point is defined by the width of the depleted region necessary to balance the potential drop induced by the work function difference. Here, the perovskite layer is semi-intrinsic, with relatively low p-type self-doping [[10.1109/JPHOTOV.2015.2451000](https://doi.org/10.1109/JPHOTOV.2015.2451000)]. Therefore, the depletion region that is close to Electron Transport Material (ETM) (e.g. PCBM) is ~ 250 - 300 nm. Only when the thickness of perovskite layer is larger than depletion width would one expect a region with flattened band profile and significantly suppressed electric field.

We expect several changes to the system such as number of potential barriers (size of conducting channel), band gap energy and crystalline packing as we increase the n value of the 2D perovskites. Keeping those parameters in mind, we conducted a thickness dependent device study with other n numbers ($n=2$ to $n=5$). The results are summarized below:

Fig R1. Thickness dependent study for other n numbered layered perovskites. a-b, average power conversion efficiency (PCE) and short circuit current density (J_{sc}) as a function of absorbing layer thicknesses for n=2 and n=5 compound respectively produced under the same processing condition. **c,** Normalized J_{sc} versus thickness plots for three different n-numbered layered perovskite devices. **d,** normalized J - V characteristics for those devices with optimized thickness (220 nm).

From the results, the thickness dependence of the device performance for all the layered system follows a similar trend. The optimized thickness for all three compounds is roughly around 200 nm. This is reasonable considering that the structures for n=2, 4, 5 compounds are similar [DOI: 10.1021/acs.chemmater.6b00847], thus rendering analogous transport properties, where the carriers collection across the stacked potential barriers is field dependent. Furthermore, from Fig. R1d, we observe a sharper decay in the case of lower n number (n=2) as compared to higher n numbers. This is consistent with our proposed mechanism where lower n-numbered thin film is more insulating likely due to a reduced ratio between conducting inorganic slab and organic spacers. This is also reflected in the normalized J - V characteristics in Fig. R1d, where a more pronounced field dependent slope from maximum power point towards short circuit can be observed in n=2 device, while the other two are more flattened.

For the electrical field profile, on the other hand, we only consider the change in distance between two electrodes and charge density distribution in this study. While, the exact field profile may differ slightly as we tune the n value, which in turn alters the dielectric constant, carrier density (absorption coefficient) and relative band alignment near the interface but importantly, the trend still holds based on the device analysis in Fig. R1.

As change to the manuscript, we have incorporated these new data as Fig. S2 in revised SI in a new section (Thickness dependence for other n-numbered RP perovskite cells) to discuss the difference in n-numbered layered perovskites, and brief discussion in the manuscript in page 6 (highlighted) refers to that.

3) What is the limit in terms of film thickness for the appearance for this flatten internal field in these layered perovskite? It would be helpful for readers to better understand the story when more explanation is provided.

This is an important aspect the reviewer has raised that needs further clarification. To accurately probe the depletion region width, we have further simulated the internal field distribution by fitting the J - V characteristics and obtained the field profile for each of the thicknesses of the perovskite film. The results are shown below for thicker devices: 300 nm and 620 nm.

Fig. R2 shows the energy band diagram for 300 nm perovskite thickness.

Fig. R3 shows the energy band diagram for 620 nm perovskite thickness.

From the results in Fig. R2 and R3, we found the depletion region width for those layered perovskite cells in this device configuration used in current study as well as our previous published study [[doi:10.1038/nature18306](https://doi.org/10.1038/nature18306)] are in the range of 200~300 nm, where internal electrical field drop uniformly across two electrodes at short circuit condition. When the thickness continues to grow (e.g. 620 nm in Fig. R3), the field tends to be flattened in the middle region of the film, beyond the depletion region near the electron contact. These data are consistent with our thickness dependent J - V curve analysis in Figure 2. In Fig. 2b in the MS, the slope near short circuit (SC) can be correlated to the charge collection efficiency under short circuit condition. This charge collection efficiency depends strongly on the drift field in the absorber layer. We find that the charge collection in the layered perovskite system begins to be field dependent above 350 nm indicated by the increased value in the J - V slope, suggesting the field strength is greatly reduced above that threshold and carriers undergo recombination before being collected. This field dependence is true for low mobility systems where carrier diffusion length is short and such upper limit may not hold for 3D perovskite system.

We therefore put these simulation data in the SI (Fig. S7-8) to show how the increase in the film thickness changes the energy band diagram and that the depletion region is in the range of 250 nm ~ 300 nm. We have also expanded the discussion paragraph in the page 10 in MS as highlighted.

4) Since the quantum wells are oriented perpendicularly respect to the substrate as depicted by GIWAXS, how the “increased number of organic spacer will intermittently disrupt the conducting pathway” as claimed in the Abstract? This seems to be contradicting.

We acknowledge the reviewer's concern and agree that if all the inorganic slabs are perfectly aligned in vertical direction, the transport between two electrodes should not be affected by the potential barriers.

However, based on the obtained device characterization and simulation results, we believe that the vertical orientation is not as perfect as in a single crystal (see Fig. R4). Instead, the packing maybe interrupted by the slight misalignment between crystalline slabs in the thin-film (Fig. R4a) as film self assembles during the hot-casting process. In contrast, the single crystal shows perfect ordering and closed packing between each crystalline slab evidenced by the sharp feature from the GIWAXS pattern (Fig. 4b). This hypothesis can be validated by comparing the single crystal diffraction pattern with the thin-film diffraction pattern as illustrated in Fig R4c-d below. The thin-film has certain degree of broadening and randomness that could be due to the discontinued/misaligned stacking along the z-axis.

Fig. R4. **a**, Cartoon illustration for the proposed misalignment in the thin film. GIWAXS maps for **b**, layered perovskite crystal and thin-film. Schematic illustration for crystal packing in **c**, single crystal and **d**, thin-film.

On the other hand, we acknowledge that such barrier could originate from other mechanisms, and the proposed schematic is just one of the possible reasons. Thus, to avoid confusion, we have revised the abstract, as well as the corresponding discussion on page 3-4 and page 18-19 of MS and add the cartoon illustration in Fig. 5i.

5) The manuscript stated that “Ruddlesden-Popper halide perovskites are two-dimensional solution-processed quantum wells and have recently emerged as highly efficient semiconductor for solar cell approaching 13% in power conversion efficiency.” However, I remember there was a paper on recent EES reported much higher efficiency as high as 13.7%. It should be corrected with proper references to show the state of the art.

We have updated the abstract and references to include this.

Reviewer #2:

This work provides a comprehensive analysis of 2D perovskite solar cells losses. The best performances are similar to the ones previously reported, and the thickness effect follows an expected trend. However, the presented results can help understanding the working mechanisms of 2D perovskite solar cells.

1) As the authors mention, the V_{oc} Vs light intensity analysis (Fig 3) suggests a dominating monomolecular recombination process at lower light intensities. Signs of the type of carrier dominating this process could be helpful to further optimization of the system.

We thank the reviewer for the insightful comment.

However, the power dependent V_{oc} measurement is a good evaluation on the type of recombination for both electron-hole [DOI: 10.1021/nm401267s]. Here in the layered perovskite system with potential barriers, we consider bimolecular processes (free carrier recombination) and monomolecular (bound carrier recombination, including bonded electron-hole, or trapped electron-hole in a potential barrier or Shockley-Read-Hall -SRH) [<https://doi.org/10.1063/1.1889240>; <https://doi.org/10.1073/pnas.1506699112>] both. During these processes, both carrier types will play a role in determining the rate of recombination that influences the performance of the solar cell. The recombination in the solar cell occurs when an electron finds a hole, independent of the specific process.

(2) In line with the previous authors' report (H. Tsai et al., Nature, 536, 7616, 312–316, 2016), which indicated a charge carrier concentration of $\sim 10^{16} \text{ cm}^{-3}$. If that is the case, the interpretation of the electric field distribution should be discussed, as it could be confined in a depletion region for films $> 200 \text{ nm}$.

We agree with the reviewer that the depletion region width might be greater than 200 nm as the charge density in the absorbing layer is low in a PIN device configuration. This measurement was done with independent experiments using capacitance characterizations for the optimized device thickness, offering a rough estimation of the depletion region width.

To accurately probe the depletion region width, we have further simulated the internal field distribution by fitting the J - V characteristics and obtained the field profile for each of the thicknesses. The results are shown below for 3 typical thicknesses: 200 nm (in Fig. S10 in revised SI), 300 nm and 620 nm.

Fig. R1 The energy band diagram for 300 nm perovskite absorber as a function of applied forward bias.

Fig. R2 The energy band diagram for 620 nm perovskite absorber as a function of applied forward bias.

From the results in Fig. R1~R2, we found the depletion region width for those layered perovskite cells in this device configuration used in current study as well as our previous published study [[doi:10.1038/nature18306](https://doi.org/10.1038/nature18306)] are in the range of 200~300 nm, where internal electrical field drops uniformly across two electrodes at short circuit condition. When the thickness continues to grow (e.g. 620 nm in Fig. R2), the field tend to be flattened in the middle region of the film, leaving depletion region near the n-type contact.

We therefore put those simulation data in the SI (Fig. S7-S8) to show how the length increase changes the energy band diagram and that the depletion region is in the range of 200-300 nm.

3) Another point which, in my opinion, deserves further analysis is the role of the electric field in the exciton dissociation mechanisms (“when the field is strong across device, the bound carriers can be separated into free carriers”). Discussing the exciton binding for this materials energy could help.

We thank the reviewer for giving us an opportunity to clarify this point. In our manuscript, the experimental data and simulation results point out that the main limiting factor for the solar cell performance in the layered perovskite system (with planar PIN configuration) is field-limited current collection. This limiting step for charge transport originates from the fact that the carriers (exciton or free) are generated inside a potential well. Independent of whether it is an exciton or free carrier, a strong electric field is necessary for the carriers to surmount the potential barrier before they recombine. In our analysis, the magnitude of the potential barrier incorporates any effects associated with the exciton binding energy. Because such a phenomenological approach is validated by the fact that the model can self-consistently explain three independent experimental data (i.e. intensity, field dependence, and temperature dependence), we can propose the mechanism without explicitly invoking the exact nature of carriers.

To further clarify our argument, we have added the appropriate discussion in page 18 in the MS.

4) The authors claim “For very thin absorbers (< 200 nm), the photo- absorption is incomplete, therefore even though the high internal field ($E \propto V_{bi}/W$) successfully collects most of the photo-generated carriers, the short circuit current is still low”. This is reasonable, but could be further confirmed with internal quantum efficiency measurements.

We agree with the reviewer that an internal quantum efficiency (IQE) measurement would validate such a statement. We have therefore measured the IQE spectra by taking the ratio of the external quantum efficiency of devices and the absorption of the full device (in reflection mode) for three different thicknesses shown in Fig. R3.

Fig. R3 Internal quantum efficiency. **a**, device absorption spectra measured in reflection mode. **b**, IQE spectra for those devices.

From the absorption spectra in Fig. R3a, we found the internal absorption increases as film thickness grows from 100 nm to 300 nm as expected. This results from the increase in film thickness, which absorbs more photons at the band edge as well as the differences in optical density with change in film thickness. This suggests that the absorption is indeed incomplete for film thinner than 200 nm.

In Fig. R3b, we found in all three cases (100 nm, 220 nm and 300 nm thick), the IQE values are above 80% and the band edge at near IR regime are much stronger than that near the visible range. This general feature is observed in many solar cells, although the exact transition depends on the specific solar cell technology. The shorter wavelength photons are absorbed close to the top contact. A fraction high energy photon (above 1.9eV or below 650 nm) are absorbed in the ITO/ETL and are lost by immediate self-recombination. The surviving photons are absorbed close to the contact. The characteristic corrugated potential makes the transport of these holes to the back-contact difficult, with the corresponding increase in the recombination. The combined effect is reflected in the IQE data.

This hypothesis is validated by the increase IQE for 100nm thin film device near 500~770 nm range (arrow in Fig. R3b) compared to the thicker device. Here the internal field is stronger, and thus the collection is more efficient. We have therefore added the IQE data in the supplementary materials Fig. S11 to support this argument, and expanded our discussion based on the result in the revised manuscript (page 16, highlighted area).

Reviewer #3:

The authors present a study on the charge transport and recombination in 2D perovskite solar cells based on quantum wells. By using JV measurements and simulations they propose a model for the workings of vertically stacked 2D perovskite solar cells. The manuscript represents a nice combination of experimental and simulation work on an important topic.

1) The manuscript contains a few typos and other language mistakes that could easily be fixed.

We thank the reviewer's careful examination on our manuscripts, we have gone through the paper carefully and corrected most typos.

2) There is no direct comparison between the experimental data and the simulations. This makes it difficult to assess how good the agreement is. Moreover, the experimental data (fig 4) don't include the FF versus T while the simulations do (fig 5). The latter, however, don't include the Voc. It would be great, if there was a direct comparison between simulations and experiments even if perfect quantitative agreement is not obtained.

We agree with the reviewer that a careful comparison between experiment and simulation would indeed validate our work. In order to do so, we have added the data (F.F. (T) plot in Fig. 4 and V_{OC} (T) in Fig. 5) as suggested by the reviewer; and also added the experimental data as grey dots to give direct comparison between experiment and theory. The new figures attached below are used as replacement in the revised MS.

Fig. R1. Revised Fig.4 in the manuscript with panel d added.

Fig. R2. Revised Fig. 5 in the manuscript with experimental data (grey color) overlay on top of simulated data taken from Fig. R1.

From the new figure, we found that the simulated trend follows the experimental data in Fig. R2 c, f, h (J_{sc} vs thickness, J_{sc} vs T and F.F. vs T respectively). These data are the key parameters describing the charge collection under field during cell operation that the model includes.

We have incorporated these changes in the manuscript and modified the text accordingly.

3) The simulations appear to be well done. However, some details are missing: In the SI it is mentioned that the optical constants are either taken from the literature (which is fine of course) or measured by the authors. In the latter case, the data should also be included in the SI (i.e. n & k for the materials). From table S1 (equations) it is not obvious to me how the offsets in the conduction and valence band (both 130 meV) are implemented. Such details are needed in order to be able to reproduce the results.

We agree with the reviewer's concern about the simulation detail. For PCBM (n,k) values are collected from Gevaerts *et al.* DOI: [10.1021/am200755m](https://doi.org/10.1021/am200755m) and for PEDOT:PSS(n,k) are collected from Hoppe *et al.* <https://doi.org/10.1080/713738799>. For 2D perovskite, we scaled the results from optical measurement by a factor k (=1.20) to match the short circuit current (as discussed in the report in [doi:10.1038/nature18306](https://doi.org/10.1038/nature18306)). No other changes in the (n, k) values were necessary to consistently interpret *all* the experimental data reported in this paper.

In order to implement the conduction and valence band offset in perovskite material, for each quantum well the electron affinity was increased by 130 meV (to simulate the offset in conduction band) and the bandgap was decreased by 260 meV (to simulate the offset in valence band) compared to values used for the parts of perovskite without quantum wells. The band-offset is not independently measured, but derived from the need to explain the temperature-dependence of charge collection. The phenomenological model is validated by the observation that the *same* offset can then self-consistently interpret all the field and illumination dependent experiments.

We will include the calculated (n, k) for the layered perovskite thin film as Fig. S12 and update the detail simulation conditions in the revised SI.

4) How are the layers implemented in the simulations? For example, how many layers are there and how thick are they? In Figs S6 and S7 the band diagrams seem to show the effect of the variations in conduction and valence band, but it is not clear whether this is actually the case. Also, in the main text, right before the summary, the authors suggest that the density of potential barriers increases with thickness? why? shouldn't the density remain constant?

We thank the reviewer for pointing these questions out and we take the opportunity to explain these points in details.

1. Quantum well in the model:

For all of the simulations (including different thicknesses), we consider four quantum wells throughout the film thickness, which gives 4 lower band gap wells along with 5 high band gap barriers. We assume the distance of those potential wells are uniformly distributed in the model. The thickness of each potential well is thus the thickness of full film divided by nine. For example, for perovskite thickness of 620 nm, 450 nm, 370 nm, 300 nm, 220 nm, and 120 nm, the thickness of quantum well is 68.6 nm, 50 nm, 41.1 nm, 33.3 nm, 24.4 nm, and 13.3 nm respectively.

2. Temperature dependence in the band structure in Fig. S6~S7 in original SI.

The variations in conduction and valence band are the way we implemented the quantum wells. We did not introduce the change in the structure into our simulation model. Figures S6 and S7 in the original SI (new Fig. S9-S10 in the revised SI) therefore show the effect of temperature on the recombination inside the quantum wells. We agree with the reviewer the exact temperature-dependence of the band gap (and band offset) is unknown and worth further investigation using

other temperature dependent spectroscopy measurements. In general, however, change in band gap energy in classical semiconductors such as silicon or GaAs is ~ 20 meV with 100 K temperature variations. Therefore, in our case, the differential temperature sensitivity of the bandgap of this temperature regions are expected to be much smaller (~ 20 meV) than the potential barrier in the materials used in this study (~ 260 meV) according to our calculation. This is also supported by the fact that the temperature dependent data from simulation by simple assumptions matches well with the experimental results. The model thus supports the observed trend in carrier transport point of view. In summary, the band structure may change under low temperature, the amplitude of E_g change may not alter our conclusion on the transport properties because the relative potential barrier that affect the transport properties remain unchanged.

3. Density of quantum well

We acknowledge the reviewer's concern on this point and this is important to clarify in our manuscript. We agree with the reviewer that for fixed n-numbered perovskite compound, the density of potential barriers (organic spacer group) should be constant, independent of the film thickness. However, the potential barriers referred to in this study arise from the spacer layers that are misaligned from the neighbor layer in vertical configuration as illustrated in Fig. R1a).

Considering the structural characterization of the layered perovskite thin film and detailed report in our previously published work ([doi:10.1038/nature18306](https://doi.org/10.1038/nature18306)), we found most of the inorganic slab are preferentially aligned in an out-of-plane configuration (illustrated in Fig. R3a). However, unlike the single crystalline sample where all the inorganic slabs are perfectly aligned throughout the sample, the thin film sample presents some degree of randomness. In order to quantitatively compare the crystalline packing, we compare the grazing incidence wide angle X-ray scattering (GIWAXS) maps for the thin film and that for single crystalline powder in Fig. R3b).

Fig. R3 Proposed crystalline packing of the layered perovskites. a, Schematic illustration of the crystalline packing. b, grazing incidence wide angle scattering (GIWAXS) maps for single

crystal and thin films. c-d crystalline structure in the crystal as compared to thin film based on the GIWAXS maps.

From the result, we observe a spot pattern in thin film and single crystal samples indicating a much superior crystalline packing as compared to that of a 3D perovskite thin film (ring structure in GIWAXS map, [doi:10.1038/srep13657](https://doi.org/10.1038/srep13657)). However, the pattern for the thin film presents a spread pattern (Fig. R3b) comparing to the sharp spots in the single crystal samples. This is because the packing in the thin film is interrupted by the loose and misalignment between crystalline slabs (Fig. R3d) compared with single crystal which has ordered and closed packing between crystalline slabs (Fig. R3 c). The thin-film has certain degree of broadening and randomness that could be the discontinued packing along z-axis, especially when film grows thicker, the number of discontinuous slab increase thus creating a higher density of potential barriers for charge extraction.

By comparing the GIWAXS of films at two different thicknesses (200 nm and 450 nm in the Fig. 2c-d in original MS), the packing remains similar suggesting that the transport behavior is intrinsic to the material. Even for the thinner device when field is still sufficient, the collection is still barrier limited evidenced by the temperature and light intensity dependent study.

In the revised manuscript, we have incorporated these changes to reflect the discussion of the above three points: 1) added the details on the simulation in the simulation section in SI; 2) added a section of discussion in the SI regarding the relative change in band gap as a function of temperature and few sentences in the main text (page 18, paragraph 1) for clarification; 3) We have modified the abstract and discussion on page 19 in the MS. We also added a new panel in Fig. 5 as Fig. 5i to visualize our model.

5) Just below fig. 4 the authors state that “As the temperature gets lower, the barrier height increases...”. I suppose the authors mean that it is increasingly more difficult to cross the barrier at lower temperatures while the barrier itself is constant. If not, could the authors clarify why the barrier depends on temperature?

The reviewer’s assumption is correct, the barrier height does not increase when the temperature decreases. Instead, the carrier transport is greatly reduced and it thus becomes harder for carriers to overcome the barrier. We have clarified the discussion in the manuscript to correct this point on page 15 paragraph 1.

Reviewers' Comments:

Reviewer #1:

Remarks to the Author:

The authors revised manuscript carefully, I am satisfied this revision.

Reviewer #2:

Remarks to the Author:

Considering the system's similarity to the previous report, I believe the main novelty of the work is the modeling of the device working mechanisms. Therefore, very strong evidence is needed to confirm the proposed hypothesis. In this aspect, the introduced modifications have improved the quality of the manuscript, and most of the simulation parameters fit well the experimental results. However, the simulated Voc trend (Fig 5g) significantly differs from the experimental values. This point, as well as the associated figure, should be further discussed.

Another important aspect of the proposed model is the internal field distribution. The authors have added simulations to support their conclusions (Figs S7 and S8). However, the process and methodology employed to calculate those are still unclear.

I would recommend further clarification on the aforementioned points prior to the manuscript publication.

Reviewer #3:

Remarks to the Author:

The authors have addressed all the issues I raised and have done so in a satisfactory manner. I recommend this manuscript be published.

Reviewer #2 (Remarks to the Author):

Considering the system's similarity to the previous report, I believe the main novelty of the work is the modeling of the device working mechanisms. Therefore, very strong evidence is needed to confirm the proposed hypothesis. In this aspect, the introduced modifications have improved the quality of the manuscript, and most of the simulation parameters fit well the experimental results.

-However, the simulated Voc trend (Fig 5g) significantly differs from the experimental values. This point, as well as the associated figure, should be further discussed.

We thank the reviewer's positive comments on our revised manuscript. We agree that most of the experimental data were well fitted by the simple distributed quantum-well based model. We are gratified to see that this simple, self-consistent model reproduced the independent experimental data such as thickness and intensity dependent JV characteristics. These are the most innovative aspect of this work. We also acknowledge that the V_{OC} (T) figure is not fitted well by the current model and it is important to discuss the difference.

The V_{OC} increases at high temperature range follows the predicted trend, but the saturation near low temperature range was not reproduced by simulation. To understand such discrepancy, we have therefore considered two more effects: a) mobility and b) band gap change with temperature.

1) Band gap change with temperature

The open circuit voltage for a planar solar cell should increase with reduced temperature in classical semiconducting systems, where the trap assisted recombination and electron-phonon interactions are greatly suppressed [doi:10.1016/j.egypro.2012.07.041]. Hybrid perovskites are unique semiconductors where the band gap energy reduces at low temperature, this is true for 3D perovskites [DOI: 10.1126/sciadv.1601156] as well as 2D perovskites [DOI: 10.1126/science.aal4211]. Based on our previous study, the optical band gap energy systematically reduces with temperature that will significantly affect the open circuit voltage at low temperature.

2) Carrier mobility

Furthermore, in a system with disordered energetic landscapes, it was proposed in organic photovoltaic [DOI: 10.1109/JPHOTOV.2016.2621346] system that the carrier mobility follows a Gaussian-disorder mobility model [DOI: 10.1103/PhysRevApplied.7.044017] of the form

$$\mu = \mu_{\infty} \exp\left(-\left(c \frac{\sigma}{kT}\right)^2\right), \quad (S1)$$

Here, μ_{∞} is asymptotic high-temperature mobility limit, σ Gaussian-disorder width, and previous work in related material has found $c \sim 2/3$ [https://doi.org/10.1002/pssb.2221750102].

Based on those two factors, we have thus refined our model in the new set of simulations, and results are summarized in Fig. R1. In the refinement, we took the temperature dependent band gap change from our previous work [DOI: 10.1126/science.aal4211]. And the Gaussian-disorder mobility has been incorporated for transport within the barrier and the quantum well regions.

With $\sigma = 50 \text{ meV}$, we find $\mu_{\infty} = 53.4, c = 0.668$ (Fig. R1b). In Fig. R1(a) shows the simulated V_{OC} (T) curve, that reproduces the saturation near low temperature range.

Fig. R1 (a) Experimental and simulated V_{OC} as a function of temperature using refined model; (b) Mobility values used in the drift diffusion simulation were fitted by the Gaussian-disorder mobility model to find μ_{∞}, σ and c .

We have therefore incorporated the new simulation in the SI as new Fig. S12, and one paragraph of discussion in the manuscript explaining the discrepancy in the V_{OC} (T) curve and possible mechanism behind it.

-Another important aspect of the proposed model is the internal field distribution. The authors have added simulations to support their conclusions (Figs S7 and S8). However, the process and methodology employed to calculate those are still unclear.

We agree with the reviewer explicitly stating the methodology to calculate the field distribution in the device is an important aspect. To enrich the description, we have included the following discussion in SI method section.

“We consider a planar solar cell geometry studied in this work (Fig. 1a), where the light incidents from the transparent electrodes with PEDOT:PSS as hole transport layer (HTL). The electric field thus decreases monotonically from HTL to the PCBM (ETM) side. The depleted width is defined as the location within the absorber where the internal field is reduced to 25% of the highest internal field at the short-circuit condition. For the cases of 220 nm and 300 nm, the perovskite layer is fully depleted based on our previous experimental results [[doi:10.1038/nature18306](https://doi.org/10.1038/nature18306)]. Fig. R2 shows a plot of the depletion width as a function of thickness of the perovskite layer. The slight peak at 380nm is related to numerical uncertainty of the simulation. The cut-off of 25% is arbitrary, other cut-off values would produce comparable results.”

These along with Fig. S7-S8 and their discussion in the SI in page 8 in section (3) fully describe the simulation details of the electrical field profiles for various film thicknesses.

Fig. R2 Depleted width in the layered perovskite device as a function of film thickness.

We have now added Fig. R2 into the revised SI as new Fig. S9.

Reviewers' Comments:

Reviewer #2:

Remarks to the Author:

The modifications have improved the quality and consistency of the work. Therefore, I would like to recommend its publication.

REVIEWERS' COMMENTS:

Reviewer #2 (Remarks to the Author):

The modifications have improved the quality and consistency of the work. Therefore, I would like to recommend its publication.

We thank the reviewer's positive response.